# Spatiotemporal single-cell regulatory atlas reveals neural crest lineage diversification and cellular function during tooth morphogenesis

Junjun Jing[1,2], Jifan Feng[1], Yuan Yuan[1], Tingwei Guo[1], Jie Lei[1], Fei Pei[1], Thach-Vu Ho[1] & Yang Chai [1] ✉

Cranial neural crest cells are an evolutionary innovation of vertebrates for craniofacial development and function, yet the mechanisms that govern the cell fate decisions of postmigratory cranial neural crest cells remain largely unknown. Using the mouse molar as a model, we perform single-cell transcriptome profiling to interrogate the cell fate diversification of postmigratory cranial neural crest cells. We reveal the landscape of transcriptional heterogeneity and define the specific cellular domains during the progression of cranial neural crest cell-derived dental lineage diversification, and find that each domain makes a specific contribution to distinct molar mesenchymal tissues. Furthermore, IGF signaling-mediated cell-cell interaction between the cellular domains highlights the pivotal role of autonomous regulation of the dental mesenchyme. Importantly, we reveal cell-type-specific gene regulatory networks in the dental mesenchyme and show that Foxp4 is indispensable for the differentiation of periodontal ligament. Our single-cell atlas provides comprehensive mechanistic insight into the cell fate diversification process of the cranial neural crest cell-derived odontogenic populations.

Craniofacial structures are highly complex tissues comprised of various cell types. The establishment of cranial neural crest cells (CNCCs) was a critical step in the evolution of the vertebrate head[1]. CNCCs are a uniquely multipotent population that give rise to numerous cell types in the vertebrate craniofacial complex, including bones and cartilage, teeth, cranial ganglia, glial cells and pigment cells[2]. Unlike trunk neural crest cells, CNCCs migrate throughout the head, differentiate into different cell types upon reaching their final destinations. Defects in the migration or differentiation of CNCCs lead to a variety of developmental abnormalities in craniofacial structures, such as cleft palate and tooth malformation[3].

Although extensive knowledge about CNCCs has accumulated over last several decades, the mechanisms that control late-stage CNCC fate decisions during craniofacial morphogenesis remain to be elucidated[4]. Recently, a single-cell study of pre-migratory and migratory neural crest cells (NCCs) revealed that early NCCs undergo a sequential series of binary cell fate decisions, and that, compared to the trunk neural crest, CNCCs have a lineage bias towards mesenchymal fate when they delaminate from the neural tube[5]. Interestingly, Zalc et al. have found that CNCC precursors undergo in vivo programming by reactivating pluripotency factors to acquire the ectomesenchymal fate[6]. Our recent study has shown that postmigratory CNCCs in the first pharyngeal arch undergo dynamic intrapharyngeal arch movement and contribute to multiple lineages as they undergo a sequence of bifurcating fate decisions[7]. Nonetheless, how these postmigratory CNCCs further diversify after acquiring mesenchymal fate

[1]Center for Craniofacial Molecular Biology, University of Southern California, Los Angeles, CA 90033, USA. [2]State Key Laboratory of Oral Diseases, National Clinical Research Center for Oral Diseases, West China Hospital of Stomatology, Chengdu, Sichuan 610041, China. ✉e-mail: ychai@usc.edu

and eventually give rise to committed cell types that form functional adult tissues remains unclear.

Gene regulatory networks (GRNs) are the information-processing system that cells utilize to govern complex spatiotemporal developmental programs[8]. Williams et al. have recently portrayed the GRN of early CNCCs by combining epigenomic and transcriptional profiling approaches[9]. They found that multiple regulatory layers including cis regulatory elements and transcription factors govern the neural crest ontogeny. However, despite advances in our knowledge of the mechanisms that control early CNCC fate decisions, limited information is available about how postmigratory CNCCs acquire cell-fate-specific programs[10]. Furthermore, the GRNs of postmigratory CNCCs at the single-cell level remain largely unknown[11].

The tooth is a vertebrate-specific organ and, similar to other ectodermal organs, including the hair and mammary gland, its development involves sequential, reciprocal epithelial–mesenchymal interactions[12]. Therefore, studies of the tooth model provide broadly applicable insights into organogenesis. Each tooth can be anatomically separated into two components: the crown which development begins during embryonic stage and the root which starts to develop postnatally[13]. Tooth development initiates with dental lamina which is derived from thickened oral epithelium and will invaginate into the underlying dental mesenchyme to form the tooth bud. The dental mesenchyme then condenses around the epithelial tooth bud and instructs tooth morphogenesis through the bud, cap, bell and crown-root transition stages[14]. The dental mesenchyme is derived from CNCCs after their migration into the oral region of first pharyngeal arch[15]; subsequently, it segregates into dental papilla and follicle lineages. Dental papilla cells give rise to dental pulp and odontoblasts, whereas dental follicle cells differentiate into periodontal tissues including the periodontal ligament (PDL), cementum, and alveolar bone[16]. The multipotency of dental mesenchyme therefore enables tooth to be an excellent model to investigate the lineage diversification of postmigratory CNCC. Moreover, the anatomy and developmental stages of tooth are well established, further underlining the competence of tooth to be an ideal model to study the late-stage CNCC. Progenitor cells in the dental mesenchyme are highly heterogeneous in vivo[17–19] and the cellular heterogeneity of mouse incisors and adult human teeth has been recently evaluated[20,21]; however, the landscapes of transcriptional heterogeneity and GRNs across mouse molar development at single-cell resolution have not yet been revealed. Furthermore, the mechanisms that control cellular domain establishment and lineage development during mouse molar formation remain unclear.

In this study, we used the mouse molar as a model to interrogate the cellular heterogeneity and GRNs that govern postmigratory CNCC fate decisions. We transcriptionally profiled single cells from mouse molars at several critical developmental stages and identified the landscape of cellular heterogeneity within the dental mesenchyme across mouse molar development from embryonic to postnatal stages. We defined the cellular domains through newly identified signature genes in the dental mesenchyme as the molar develops and found that each one makes a specific contribution to mouse molar formation. Furthermore, we investigated the cell-cell interaction in the dental mesenchyme and found that disruption of IGF signaling-mediated cell–cell interaction disturbed PDL development, highlighting the importance of cell-autonomous regulation for postmigratory CNCCs. Importantly, we identified cell population-specific regulons in the dental mesenchyme and found that loss of a critical regulon, Foxp4, can lead to developmental defects in PDL lineage. Thus, this single-cell atlas reveals previously unidentified heterogeneity within the mesenchyme of the developing molar and provides comprehensive understanding of the mechanisms governing CNCC-derived cell lineage diversification during molar morphogenesis.

## Results

### Postmigratory CNCC segregation and lineage commitment during early tooth development

Postmigratory CNCCs commit to the dental mesenchymal lineage after they arrive in the oral region of the first pharyngeal arch. The dental tissues then further differentiate through a series of developmental steps until the tooth erupts. To comprehensively reveal the cellular heterogeneity and function of postmigratory CNCCs during mouse molar development, we sequenced the individual cell transcriptomes in the tooth and surrounding tissue from E13.5 to P7.5 (Fig. 1a). We first analyzed the cell populations at E13.5, the onset of tooth development.

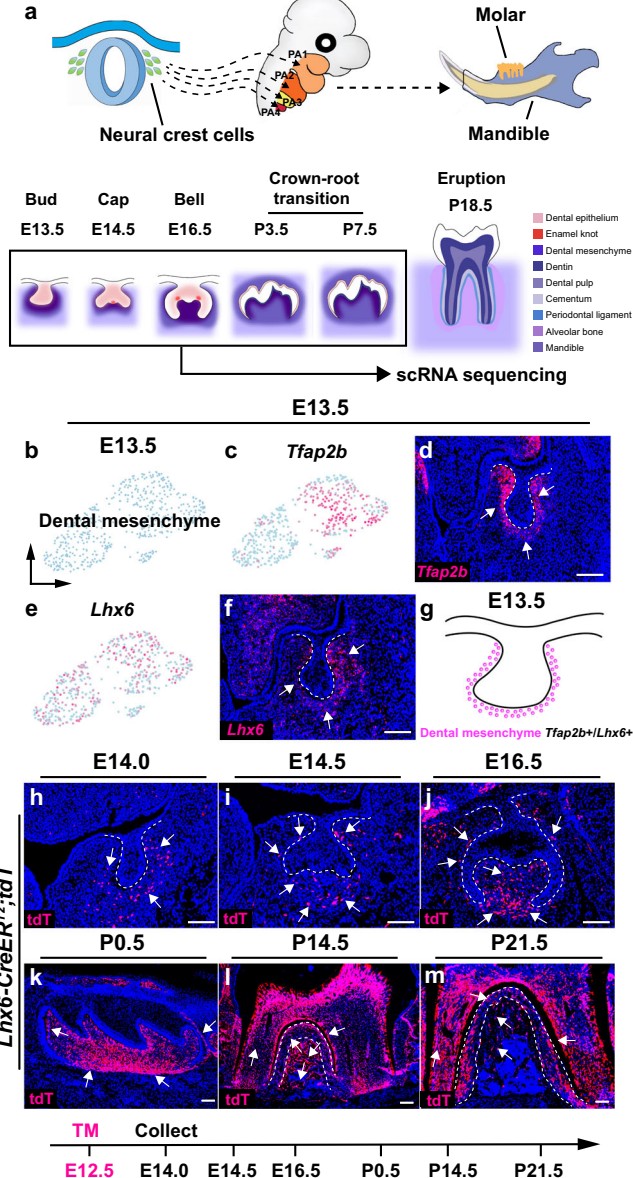

**Fig. 1 | Dental mesenchymal cell populations are relatively homogenous at E13.5. a** Schematic drawings illustrating the experimental design in this study. **b** UMAP plot of the cell cluster of the dental mesenchyme in the mouse molar at E13.5. **c–f** Feature plot and RNAscope staining of *Tfap2b* and *Lhx6* in the mouse molar at E13.5 (*n* = 3). **g** Schematic drawing of the cellular domain in the dental mesenchyme of the mouse molar at E13.5. **h–m** Lineage tracing experiment with *Lhx6-CreER^T2^;tdT* mouse molars (*n* = 3). Injection protocol is illustrated underneath the figures. White dotted lines in **d**, **f**, **h**, **i** and **j** outline the dental epithelial cells and in **l** and **m** outline the PDL in the mouse molar. White arrows point to the positive signals. Scale bars, 100 μm.

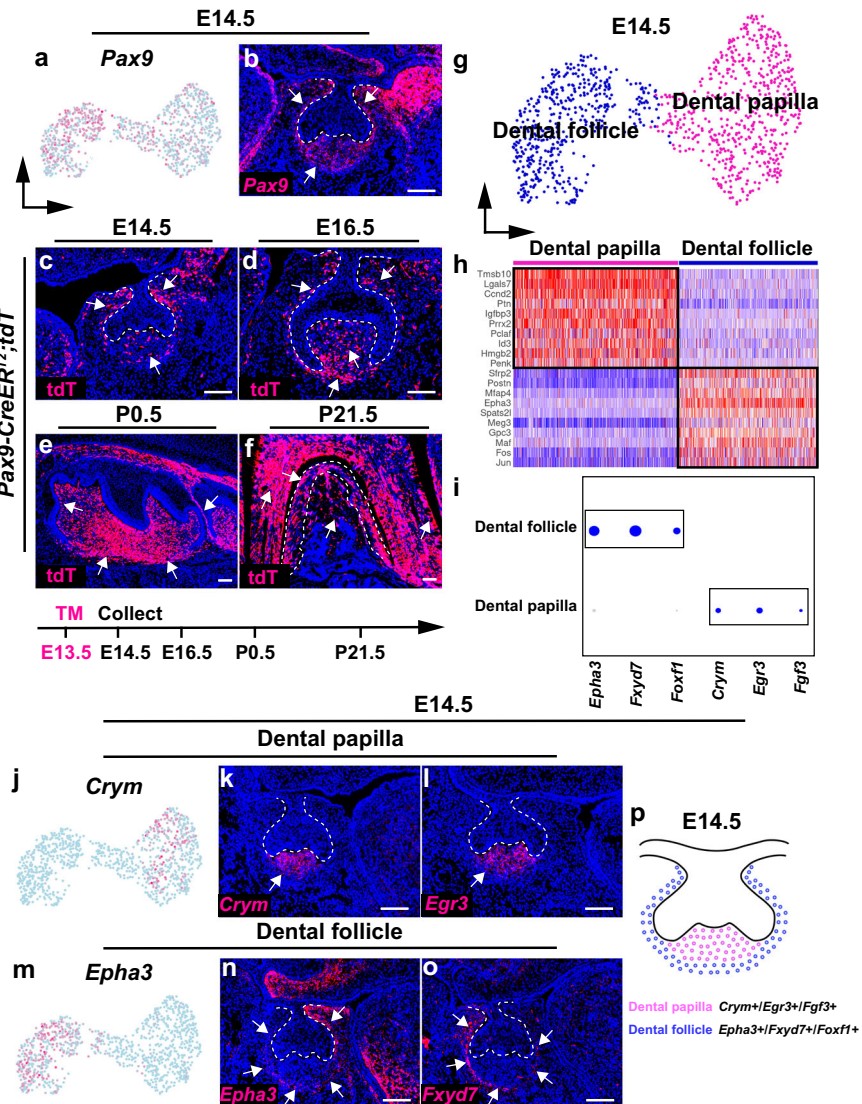

**Fig. 2 | Dental follicle and dental papilla lineages segregate at E14.5. a, b** Feature plot and RNAscope staining of *Pax9* in the mouse molar at E14.5 (*n* = 3). **c–f** Lineage tracing analysis of Pax9+ cells in the dental mesenchyme of the mouse molar from E14.5 to P21.5 (*n* = 3). **g** UMAP plot of the dental mesenchyme in the mouse molar at E14.5. **h, i** Heatmap and Dotplot of signature genes in the dental papilla and follicle of the mouse molar at E14.5. **j–l** Feature plot and RNAscope staining of dental papilla markers in the mouse molar at E14.5 (*n* = 3). **m–o** Feature plot and RNAscope staining of dental follicle markers of the mouse molar at E14.5 (*n* = 3). **p** Schematic drawing of the cellular domain in the dental mesenchyme of the mouse molar at E14.5. White dotted lines in **b, c, d, k, l, n** and **o** outline the epithelial cells and in **f** outline the PDL in the mouse molar. White arrows point to the positive signals. Scale bars, 100 μm.

Leveraging unsupervised clustering and marker analysis, we identified a large group of CNCC-derived cells which include chondrogenic cells, osteogenic cells, dermal fibroblasts and dental mesenchyme (Supplementary Fig. 1a–c). Through further subclustering analysis with a known early dental mesenchyme marker, Tfap2b[22], one cell population (*Tfap2b+/Lhx6+*) was identified as the dental mesenchyme at E13.5 (Fig. 1b), suggesting that dental mesenchymal cells are relatively homogenous right after postmigratory CNCCs commit to the dental fate at E13.5. Consistent with marker analysis, RNAscope in situ staining of these marker genes showed they are expressed in the dental mesenchyme surrounding the dental epithelium (Fig. 1c–f). The cellular domain of the dental mesenchyme in the mouse molar at E13.5 is illustrated in Fig. 1g. To confirm that this cell population we identified at E13.5 is the progenitor population that contributes to the tooth formation, we performed lineage tracing experiments with *Lhx6-CreER^{T2};tdT* mice. Tamoxifen was injected at E12.5 to enable the activation of CreER[T2] and we verified that Lhx6+ cells are present in the dental mesenchyme at E14.0 (Fig. 1h). We found that Lhx6+ cells can

give rise to all the lineages in the dental mesenchyme by P21.5 (Fig. 1i–m), suggesting that the Lhx6+ cell population at E13.5 is indeed the dental mesenchymal progenitor population that contributes to tooth formation.

In order to interrogate the lineage development of dental mesenchymal cells, we further analyzed the cell populations in the mouse molar at E14.5. Utilizing known dental mesenchyme markers at the cap stage, Tfap2b and Pax9[23], we were able to pinpoint the dental mesenchymal cell populations more specifically at this stage (Fig. 2a, b and Supplementary Fig. 2a–c). Consistent with previous studies, we found that *Pax9* is expressed in both the dental follicle and papilla at E14.5. To confirm that Pax9+ cells in the mouse molar at E14.5 are dental mesenchymal progenitor cells, we performed lineage tracing experiments with *Pax9-CreER^{T2};tdT* mice induced with tamoxifen at E13.5. The results clearly showed that Pax9+ cells are located in the dental mesenchyme at E14.5 and they are indeed progenitor cells, because they can give rise to all the mesenchymal lineages in the mouse molar at P21.5 (Fig. 2c–f). Through subclustering analysis of

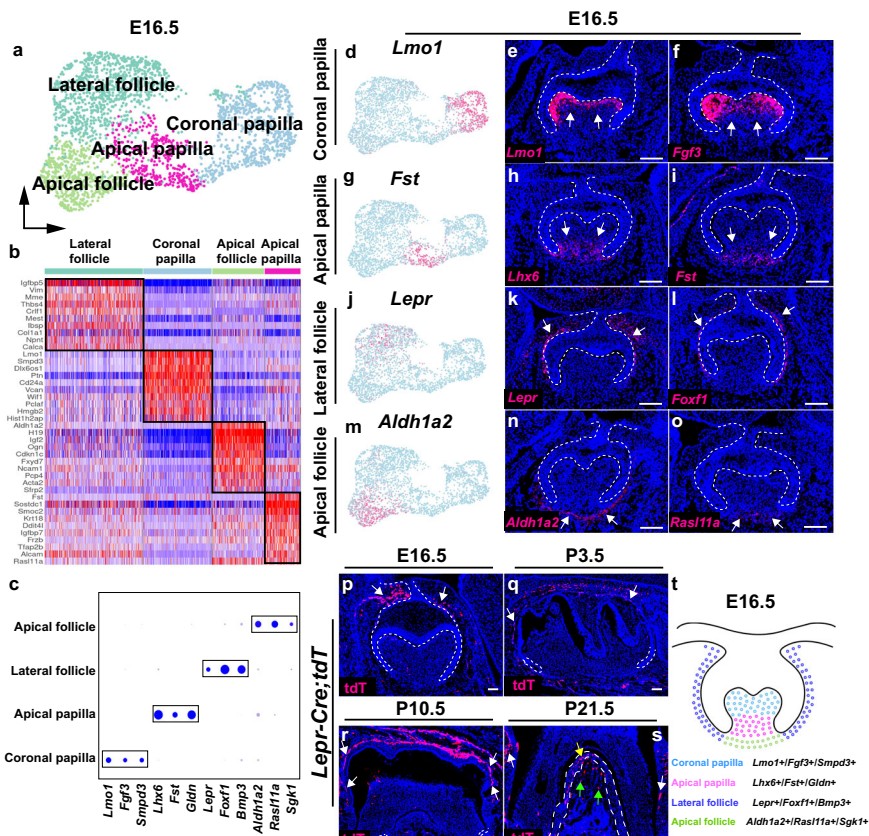

**Fig. 3 | Dental cell populations evolve into four distinctive cellular domains at E16.5. a** UMAP plot of cell clusters in the dental mesenchyme of the mouse molar at E16.5. **b**, **c** Heatmap and Dotplot of signature genes in the dental papilla and follicle of the mouse molar at E16.5. **d–o** Feature plot and RNAscope staining of dental papilla and dental follicle markers in the mouse molar at E16.5 (*n* = 3). **p–s** Lineage tracing experiment performed with *Lepr-Cre;tdT* mice (*n* = 3). **t** Schematic drawing of the cellular domain in the dental mesenchyme of the mouse molar at E16.5. White dotted lines outline the dental epithelial cells and PDL in the mouse molar. White arrows point to the positive signals. Yellow and green arrows in **s** point to the positive signals in PDL and alveolar bone in the root furcation region, respectively. Scale bars, 100 μm.

Pax9+ cells, we found two well-separated clusters corresponding to dental follicle and dental papilla cell populations, suggesting these two lineages segregate at E14.5 (Fig. 2g and Supplementary Fig. 3a–f). We identified the top enriched genes in these two cell populations and found previously unknown unique markers for these two lineages with unbiased marker analysis (Fig. 2h, i). Through RNAscope in situ staining, we found dental papilla marker genes (*Crym+/Egr3+/Fgf3+*) that are specifically expressed in the dental papilla and absent from the dental follicle (Fig. 2j–l and Supplementary Fig. 3a–c). Conversely, we found dental follicle marker genes (*Epha3+/Fxyd7+/Foxf1+*) that are expressed in the dental follicle cell domain surrounding the dental epithelium and dental papilla (Fig. 2m–o and Supplementary Fig. 3d–f). These two distinct cellular domains of the dental papilla and follicle in the mouse molar at E14.5 are illustrated in Fig. 2p.

**Evolution and spatial separation of cellular domains in the mouse molar**

At E16.5, the mouse molar reaches the bell stage of development, at which the crown patterning is determined. Through subclustering and unsupervised marker analysis, we were able to identify four cell populations in the dental mesenchyme at E16.5 (Fig. 3a, Supplementary Fig. 4a–c and Supplementary Fig. 5a–l). Varied signature genes were revealed among these cell clusters (Fig. 3b, c). At this stage, the dental papilla cell population has formed separate coronal (*Lmo1/ Fgf3/Smpd3+*) and apical domains (*Lhx6+/Fst+/Gldn+*) (Fig. 3d–i and Supplementary Fig. 5a–f), a reflection of further differentiation of the dental papilla lineage compared to E14.5. Likewise, the dental follicle cells have also evolved into two cellular domains which are located

in the lateral (*Lepr+/Foxf1+/Bmp3+*) and apical regions (*Aldh1a2+/ Rasl11a+/Sgk1+*) of the dental follicle, respectively (Fig. 3j–o and Supplementary Fig. 5g–l), implying they contribute to the development of spatially distinct regions within the tooth. In order to validate the contributions of the dental follicle cell populations, we generated *Lepr-Cre;tdT* mice to investigate the contribution of Lepr+ lateral follicle cells during mouse molar development. Lepr+ cells have been reported to be bone marrow mesenchymal stem cells in the long bone[24]. We found Lepr+ cells in the dental follicle can give rise to the periodontal tissue including PDL and alveolar bone, but these cells were absent from the dental papilla-derived dental pulp and odontoblasts in the root region of the mouse molar, indicating Lepr+ cells are indeed dental follicle progenitor cells that contribute to development of the periodontal tissue in the mouse molar (Fig. 3p–s). The cellular domains of the dental papilla and follicle in the mouse molar at E16.5 are illustrated in Fig. 3t.

To better understand the cellular heterogeneity and role of dental mesenchyme during mouse molar root development, we isolated cells from the molar at P3.5, at which time root development is about to initiate. Six cell populations were assigned to the dental mesenchyme according to the unsupervised clustering and newly identified signature gene analysis at P3.5 (Fig. 4a–c and Supplementary Fig. 6). A lateral follicle domain (*Bmp3+/Tnmd+*) and an apical follicle domain (*Smoc2+/Slc1a3+*) were defined based on the expression patterns of these newly identified signature genes at P3.5 (Fig. 4d–k). The follicle pattern appeared similar to that observed at E16.5, suggesting that the cellular domains in the dental follicle are already established at E16.5. In order to examine the contribution of dental follicle cells to tooth

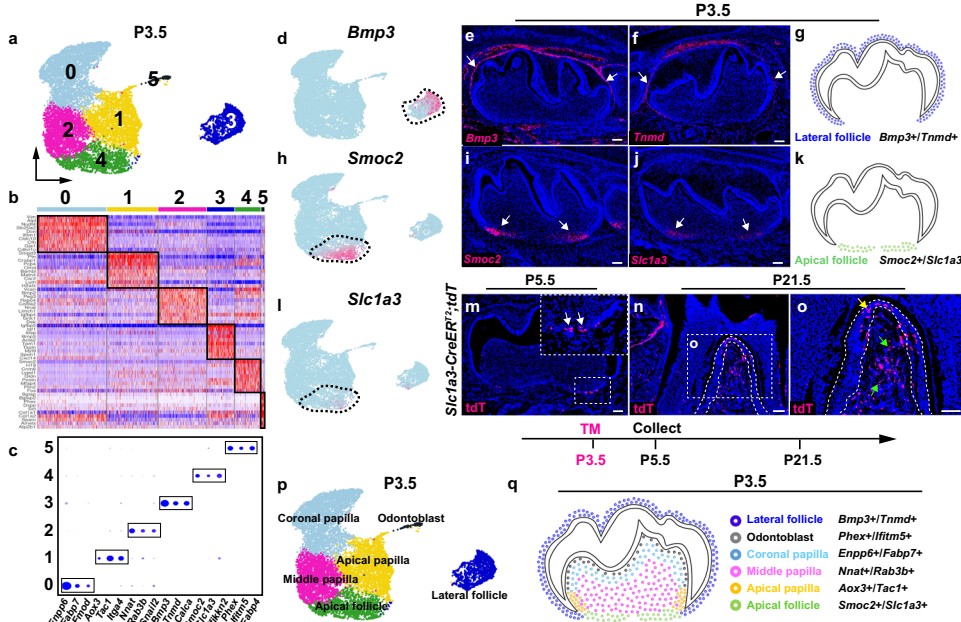

**Fig. 4 | Apical dental follicle cells largely contribute to the periodontal tissue in the furcation region of the mouse molar. a** UMAP plot of cellular domains in the mouse molar at P3.5. **b, c** Heatmap and Dotplot of signature genes of dental mesenchymal cell populations in the mouse molar. **d−f** Feature plot and RNAscope staining of lateral follicle markers in the mouse molar at P3.5 (*n* = 3). **g** Schematic drawing of the lateral follicle domain in the mouse molar at P3.5. **h−j** Feature plot and RNAscope staining of apical follicle markers in the mouse molar at P3.5 (*n* = 3). **k** Schematic drawing of the apical follicle domain in the mouse molar at P3.5. **l−o** Feature plot of *Slc1a3* and lineage tracing analysis of Slc1a3+ cells in the mouse

root from P3.5 to P21.5 (*n* = 3). **p** UMAP plot of cellular domains in the mouse molar at P3.5 with annotation. **q** Schematic drawing of the cellular domains in the mouse molar at P3.5. Injection protocol is illustrated underneath the figures. Black dotted lines outline the cellular domains in the Feature plots. The white box in **n** is magnified in **o**. White dotted lines in **o** outline the PDL in the mouse molar. White arrows point to the positive signals. Yellow and green arrows in **o** point to the positive signals in PDL and alveolar bone in the root furcation region, respectively. Scale bars, 100 μm.

root development at P3.5, we first analyzed the expression of *Pthlh* (the gene encoding Pthrp), a known marker of the dental follicle in the mouse molar[17], and confirmed it is expressed in both the lateral and apical follicle domains (Supplementary Fig. 7a, b). To validate the contribution of Pthrp+ cells, we performed a lineage tracing experiment with *Pthrp-CreER^T2;tdT* mice. Tamoxifen was injected at P3.5 to enable the activation of CreER^T2 and, consistent with its expression pattern, we found Pthrp+ cells in both the lateral and apical domains of the dental follicle two days after the tamoxifen induction (Supplementary Fig. 7c). We also revealed that Pthrp+ cells can give rise to periodontal tissue in both the furcation and root-forming regions of the mouse molar (Supplementary Fig. 7d, e). To further understand the contributions of the apical domains in the dental follicle, we generated *Slc1a3-CreER^T2;tdT* mice to target the apical follicle domain and performed lineage tracing. Slc1a3 is a known marker for astrocytes in the nervous system[25]. The cell lineage tracing data showed that Slc1a3+ cells are located in the apical region of the dental follicle two days after tamoxifen induction and that they largely contribute to the PDL and alveolar bone formation in the root furcation region where the two roots of mouse molar are separated (Fig. 4m−o), suggesting dental follicle cells in the apical domain govern the cell lineages for root furcation development. The cellular domains of the dental mesenchyme at P3.5 were illustrated in Fig. 4q based on the analysis of the expression of signature genes and lineage tracing data (Fig. 4d−p and Supplementary Fig. 8).

### Postnatal apical papilla domain retains bipotent progenitor cells that give rise to dental pulp cells and odontoblasts during mouse molar development

In analyzing the cellular heterogeneity within the CNCC-derived dental papilla, we identified four cellular domains with newly identified marker genes in the dental papilla of the mouse molar at P3.5 through

unsupervised marker analysis. Odontoblasts (*Phex*+/*Ifitm5*+) are terminally differentiated dentin-forming cells that are located underneath the dentin in the mouse molar (Supplementary Fig. 8a-d). Cell populations within the coronal dental papilla (*Enpp6*+/*Fabp7*+), middle dental papilla (*Nnat*+/*Rab3b*+) and apical dental papilla (*Aox3*+/*Tac1*+) are dental papilla cells with different differentiation status (Supplementary Fig. 8e−p). In order to understand the differentiation trajectory of these cell populations in the dental papilla, we took advantage of Velocity analysis, which infers the differentiation status of the cells based on the ratio of spliced and unspliced mRNA[26]. The results indicated that apical dental papilla cells are progenitor cells that can give rise to odontoblasts and the dental pulp lineage (Fig. 5a). Because highly proliferative cells are located in the apical region of the dental papilla in the mouse molar (Fig. 5b, c), in order to confirm the differentiation trajectory in vivo, we first examined whether EdU+ cycling cells can contribute to the dental pulp and odontoblast formation through an EdU tracing experiment. The results suggested that EdU+ cells within the apical dental papilla domain indeed give rise to odontoblast and pulp lineages nine days after tracing (Fig. 5d, e). In addition, we found that *Fgf3*, encoding a ligand for FGF signaling[27], is specifically expressed in the apical dental papilla at P3.5 (Fig. 5f, g). To confirm whether Fgf3+ cells in the apical papilla are the progenitor cells for dental pulp and odontoblast formation, we generated *Fgf3-CreER^T2;tdT* mice to enable tracking Fgf3+ cells in the mouse molar at P3.5. The results showed that Fgf3+ cells in the apical papilla can become dental pulp cells and odontoblasts (Fig. 5h, i), suggesting apical dental papilla cells are the bipotent progenitor cells that generate pulp and odontoblast lineages in the mouse molar (Fig. 5j). Dental pulp cells in the coronal region have been reported to form odontoblasts during post-injury repair[28,29], indicating the mechanisms that regulate the differentiation trajectory of dental papilla progenitors are different in developmental and repair contexts.

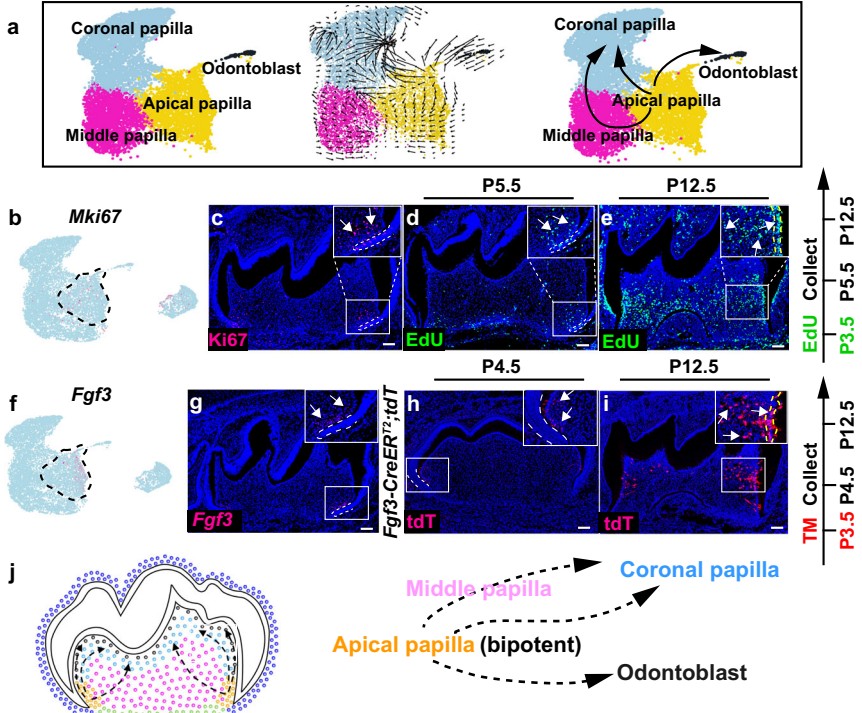

**Fig. 5 | Apical papilla cells are bipotent progenitor cells that give rise to dental pulp cells and odontoblasts in the mouse molar. a** Velocity analysis of dental papilla cell populations in the mouse molar at P3.5. **b, c** Feature plot of *Mki67* and immunostaining of Ki67 in the mouse molar at P3.5 (*n* = 3). **d, e** EdU tracing analysis in the mouse molar from P3.5 to P12.5 (*n* = 3). **f, g** Feature plot and RNAscope staining of *Fgf3* in the mouse molar at P3.5 (*n* = 3). **h, i** Lineage tracing analysis of Fgf3+ cells in the mouse molar from P3.5 to P12.5 (*n* = 3). **j** Differentiation trajectory of dental papilla cells in the mouse molar at P3.5. White boxes are magnified in the inserts of the figures. Black dotted lines outline the cellular domains in the feature plots and white dotted lines outline the dental epithelial cells in the mouse molar. Yellow dotted lines in **e** and **i** outline the odontoblast layer in the mouse molar. White arrows point to the positive signals and black arrows indicate the differentiation trajectory. EdU and tamoxifen injection protocols are on the right side of the figures. Scale bars, 100 μm.

## Different sets of genes switch on and off along the differentiation trajectory of the dental mesenchymal lineages during mouse molar development

To investigate whether the cellular domain for tooth root development is fully developed at P3.5, we further analyzed the cell populations in the mouse molar at P7.5 and found that the cellular domains and corresponding signature genes in the dental mesenchyme at this stage resembled those at P3.5 (Supplementary Fig. 9), suggesting the cellular domains for tooth root formation are established at P3.5. In order to understand the subtle change in the genetic programs that determine cell fate along the differentiation trajectory of the dental mesenchymal lineages during mouse molar development, we first performed integrative analysis of the dental papilla lineage single-cell RNA-seq data from five developmental stages mentioned above (E13.5, E14.5, E16.5, P3.5, P7.5) using Seurat 4. We identified the cell populations that can be categorized into dental pulp and odontoblast lineages based on their highly specific marker genes (Fig. 6a, b). To investigate the dental papilla lineage development, we first used scVelo to perform Velocity analysis[25]. We found that, consistent with our in vivo discovery, the common progenitor cells divided into two major branches, corresponding to the aforementioned dental pulp and odontoblast lineages (Fig. 6c). Then Monocle 3 pseudotime analysis, which provided a computational model of the cell fate decisions of the dental papilla cells[30], was conducted to confirm this finding (Fig. 6d–i). The earliest dental mesenchyme marker identified in this study, Tfap2b, was used to refer to the root node in the pseudotime analysis. To further understand the cell fate transition of the progenitor cells in the dental papilla, we performed GeneSwitches analysis, through which we identified the sets of genes that are switched on/off in particular lineages[31]. Interestingly, many marker genes identified in each lineage are switched "on" in those specific lineages, providing another layer of evidence regarding the cell lineage development during mouse molar formation (Fig. 6j, k). Likewise, we performed similar analysis of the dental follicle lineage and, consistent with what we found in vivo, the progenitor cells give rise to cell populations in the apical and lateral follicle along the differentiation trajectory (Fig. 6l–t). Different sets of genes were also found to be switched on/off during the progression of dental follicle lineage development, highlighting the dynamic change of the genetic programs governing the cell fate transition of the progenitor cells in the dental mesenchyme (Fig. 6u, v).

## Cell–cell interaction between the cellular domains in the dental mesenchyme is important for lineage development in the mouse molar

To gain insights into potential signaling interactions between different cellular domains in the dental mesenchyme, we chose to analyze the dataset from P3.5 to be representative. We interrogated our data with the CellChat package[32], which enables the prediction of ligand and receptor interactions at single-cell resolution. Multi-directional signaling interactions between the cellular domains at P3.5 are depicted in Fig. 7a, highlighting the sophisticated interactions between the dental mesenchymal cell populations. To further elucidate the interaction pattern, we analyzed the outgoing and incoming signaling patterns and the data suggested that each cellular domain serves as a source for different signaling ligands (Fig. 7b and Supplementary Fig. 10). For instance, the apical papilla and odontoblasts secrete Wnt10a, a typical canonical WNT ligand. Previous studies demonstrated that loss of canonical WNT signaling in the dental mesenchyme leads to a tooth root development defect[33]. The middle papilla and apical follicle serve as a source of Bmp2, and we have previously shown that disruption of BMP signaling in tooth root progenitor cells results in the absence of root formation[18].

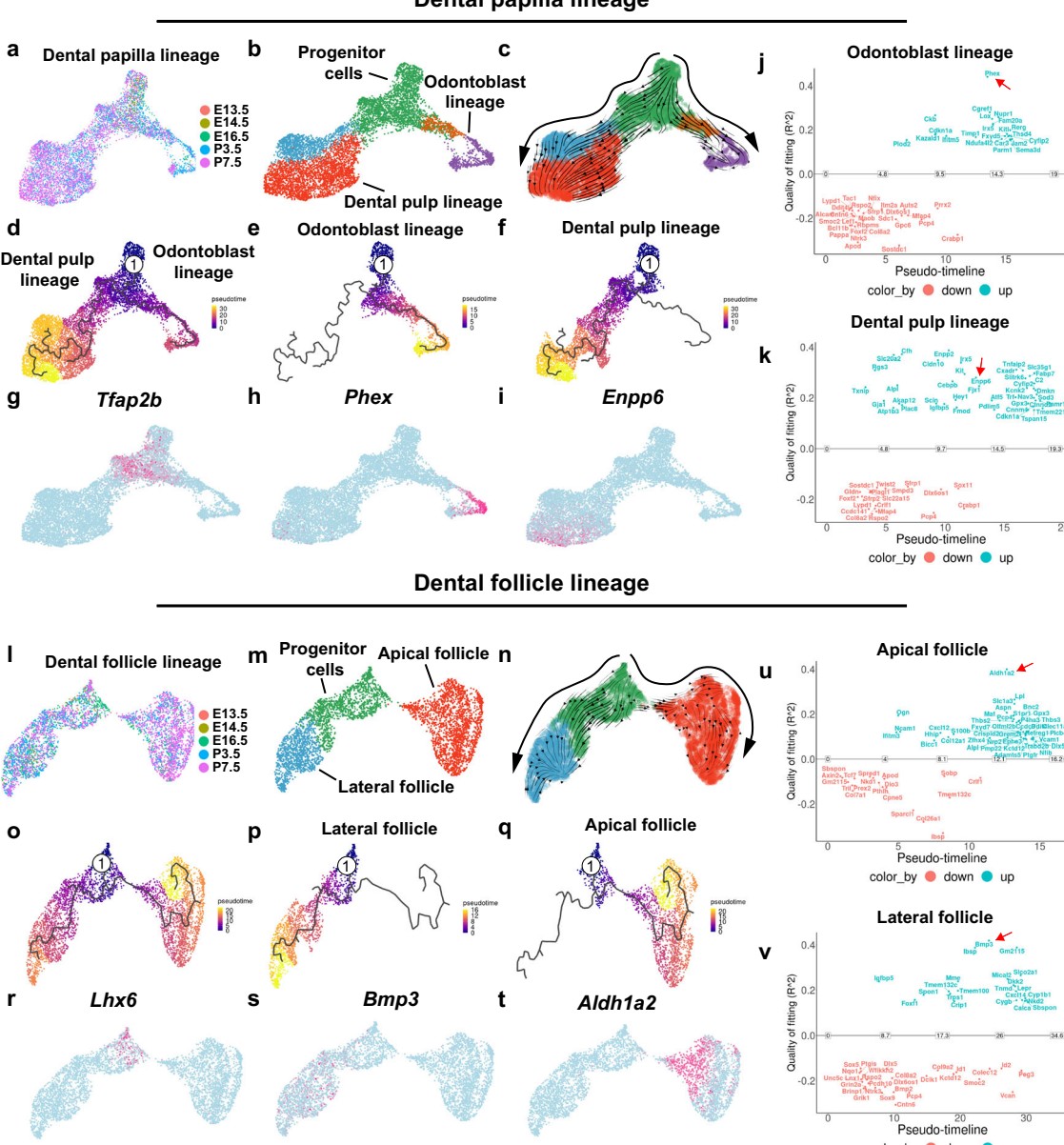

**Fig. 6 | Differentiation trajectory and lineage transition in the dental mesenchyme. a** Integrative analysis of the dental papilla lineage data from E13.5, E14.5. E16.5, P3.5 and P7.5. **b** Dental papilla clusters in integrative analysis. **c** Differentiation trajectory inference of dental papilla lineage with Velocity analysis. **d–f** Differentiation trajectory inference of dental papilla lineage with Monocle analysis. **g–i** Feature plot of *Tfap2b*, *Phex* and *Enpp6* from the dental papilla lineage. **j, k** Odontoblast and dental pulp lineage inference with GeneSwitches analysis. Red arrows highlight *Phex* and *Enpp6* in **j, k. l** Integrative analysis of the dental follicle lineage data from E13.5, E14.5. E16.5, P3.5 and P7.5. **m** Dental follicle clusters in integrative analysis. **n** Differentiation trajectory inference of dental follicle lineage with Velocity analysis. **o–q** Dental follicle lineage inference with Monocle analysis. **r–t** Feature plot of *Lhx6*, *Bmp3* and *Aldh1a2*. **u, v** Apical and lateral follicle inference with GeneSwitches analysis. Red arrows highlight *Aldh1a2* and *Bmp3* in **u, v.** Number 1 indicates the root node in the trajectory analysis.

IGF signaling was predicted to mediate the interaction between lateral follicle cells and other cellular domains (Fig. 7b). To validate the expression pattern of IGF ligands, RNAscope in situ staining of *Igf1* and *Igf2* was conducted in the mouse molar at P3.5. The data showed *Igf1* is specifically expressed in the dental follicle whereas the expression of *Igf2* is almost undetectable in the dental mesenchyme. *Igf1r* is widely expressed in the mouse molar, suggesting that the dental follicle serves as the source of Igf ligands to activate IGF signaling via Igf1r in the cellular domains of the dental mesenchyme (Fig. 7c–h). To test the function of IGF signaling-mediated cell-cell interaction in the dental mesenchyme, we generated *Lepr-Cre; Igf1*^fl/fl and *Slc1a3-CreER*^T2;*Igf1r*^fl/fl mouse models in which *Igf1* and *Igf1r* were deleted in the lateral and apical follicle, respectively, in the mouse molar. We found that loss of either *Igf1* or *Igf1r* has no apparent impact on the root length of the mouse molar whereas the PDL area is larger in both mutants than controls at P16.5, suggesting relatively undifferentiated status of PDL in the mutants (Supplementary Fig. 11a, b). Compromised PDL differentiation in both mutant models was further confirmed by reduced expression of periostin, a marker of PDL differentiation (Fig. 7i–q), suggesting Igf1-Igf1r mediated cell-cell interaction between lateral and apical follicle domains is important for PDL development.

**Gene regulatory networks determine the lineage development in the dental mesenchyme during mouse molar morphogenesis**
GRNs play key roles in controlling cell fate specification and differentiation[34]. To uncover the mechanisms that govern the lineage

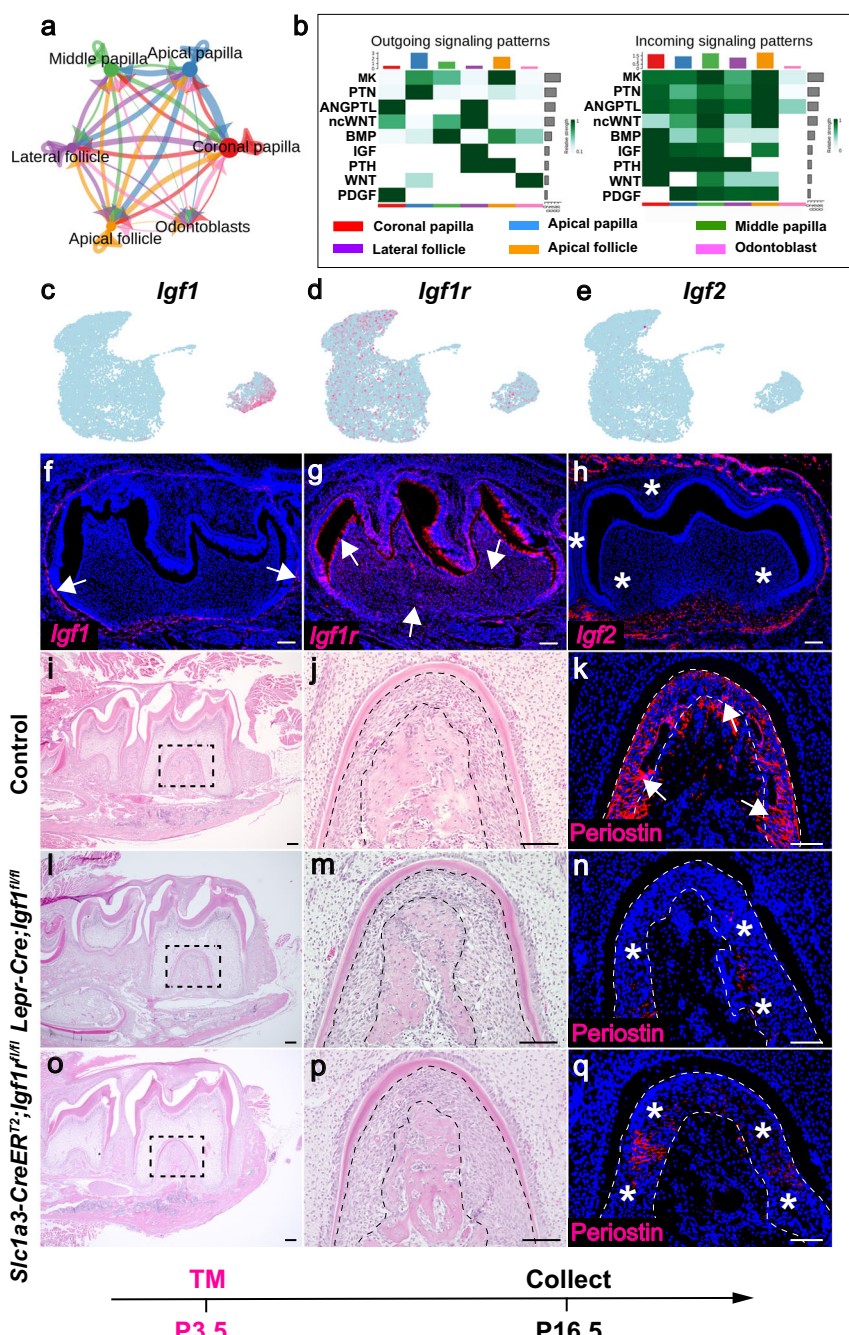

**Fig. 7 | Cell–cell interaction in the dental mesenchyme is important for the lineage development of the mouse molar. a** NetVisual_circle plot of the number of interactions between the cellular domains in the dental mesenchyme at P3.5. **b** Outgoing and incoming signaling patterns between the dental mesenchymal cell populations P3.5. **c–h** Feature plot and RNAscope staining of *Igf1*, *Igf1r* and *Igf2* in the mouse molar at P3.5 (*n* = 3). **i–q** H&E staining and immunostaining of periostin in the mouse molar in control, *Lepr-Cre;Igf1^fl/fl^* and *Slc1a3-CreER^T2^;Igf1r^fl/fl^* mice at P16.5 (*n* = 3). The black and white dotted lines outline the PDL in the mouse molar. The white arrows point to the positive signals and the asterisks indicate absence of the signal. Injection protocol is illustrated underneath the figures. Scale bars, 100 μm.

development in the dental mesenchyme, we applied a SCENIC workflow for identifying cell-type-specific regulons to comprehensively reconstruct the GRNs in the dental mesenchyme (Fig. 8a)[35]. We first identified cell-type-specific regulons in the dental mesenchyme at E14.5 when the dental papilla and follicle lineages segregate (Fig. 8b). We found that several transcription factor families are involved in these two distinct lineages. For example, Creb family members including Creb1, Creb3 and Creb5 are associated with the dental follicle, whereas Elf family members such as Elf1, Elf2 and Elf4 are associated with the dental papilla. The regulons highlighted at E14.5

revealed that the predicted transcription factors (TFs) and corresponding target genes were expressed in the same cellular domain, suggesting that GRNs enriched in the dental papilla and follicle are unique and important for the cell fate determination of these populations (Supplementary Fig. 12a, b). To investigate the mechanisms that maintain the identity of dental mesenchymal cell populations in the mouse molar at E16.5, we performed regulon analysis in the dental mesenchyme at this stage (Fig. 8c). We found that, consistent with previous findings, several transcription factor families, such as Dlx family members Dlx2 and Dlx5, are highly enriched in the dental papilla

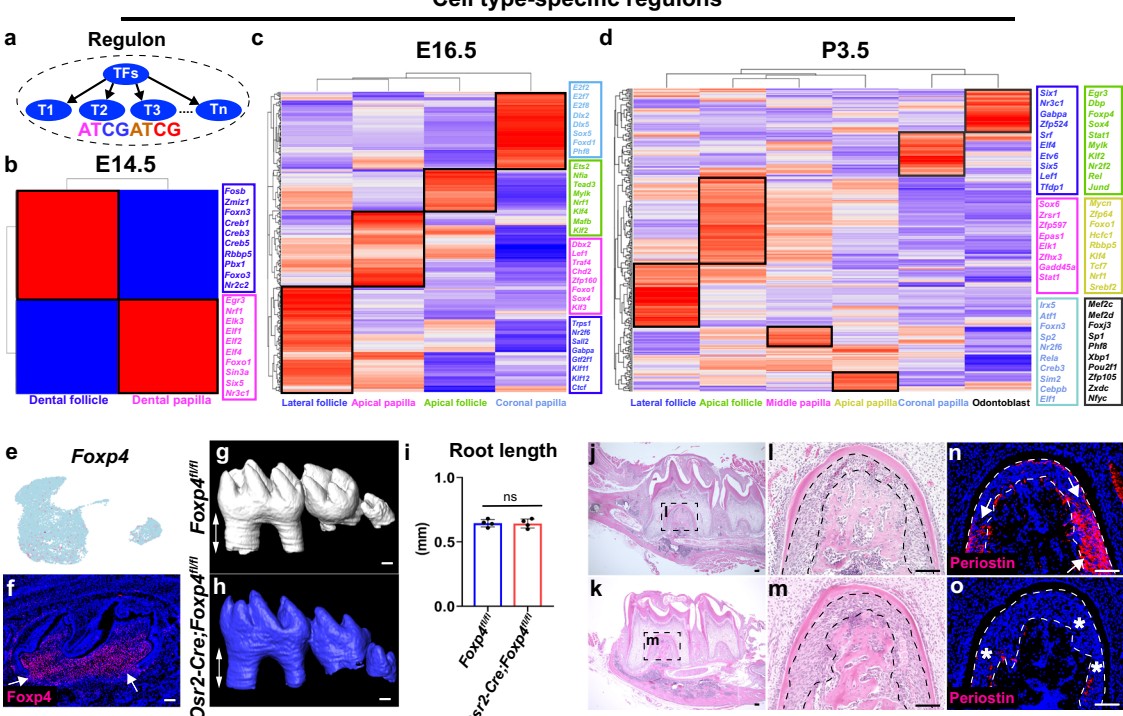

**Fig. 8 | Gene regulatory networks determine dental mesenchymal lineages development during mouse molar formation. a** Schematic drawing of a regulon, TFs, transcription factors, T, target gene. **b–d** Heatmap of cell type-specific regulons and selected top enriched regulons in the cell populations of the dental mesenchyme at E14.5, E16.5 and P3.5. **e, f** Feature plot and immunostaining of Foxp4 in the mouse molar at P3.5 (n = 3). **g–i** CT scanning and quantitative analysis of the tooth root length in *Foxp4^{fl/fl}* and *Osr2-Cre;Foxp4^{fl/fl}* molars at P16.5, ns, no significant difference. Data are presented as mean values ± SD. n = 4 biologically independent samples, two-tailed t test. **j-o** H&E staining and immunostaining of periostin in the molars of *Foxp4^{fl/fl}* and *Osr2-Cre;Foxp4^{fl/fl}* mice at P16.5 (n = 3). Black and white dotted lines outline the PDL in the mouse molar. The white arrows point to the positive signals and the asterisks indicate absence of the signal. Scale bars in CT images are 200 μm, in other images 100 μm.

cell populations[36], whereas Klf family members such as Klf4 and Klf11 are highly enriched in the dental follicle populations[24,37]. GRNs revealed with co-expressed TFs and target genes suggested they are involved in regulating dental mesenchymal lineage development in the mouse molar at E16.5 (Supplementary Fig. 12c, d).

To reveal the function of GRNs in regulating the cell lineage development during molar root formation, we conducted regulon analysis with the SCENIC pipeline on the P3.5 dataset and identified cell-type-specific regulons in the dental mesenchyme which are potentially important to maintain cellular identity and drive cellular heterogeneity in the molar mesenchyme at P3.5 (Fig. 8d and Supplementary Fig. 12e, f). In particular, FOX (forkhead box) members, such as Foxp4, Foxn3, and Foxj3, are highly enriched in the cellular domains of the mouse molar at P3.5, indicating that FOX transcription factors are widely involved in the regulation of dental mesenchymal cells at P3.5. This is consistent with the importance of the FOX transcription factor superfamily in both tissue development and homeostasis[38].

Consistent with marker analysis, Foxp4 is expressed in both the dental papilla and follicle of the molar at P3.5 (Fig. 8e, f). To validate the function of Foxp4 in tooth development, we generated *Osr2-Cre;-Foxp4^{fl/fl}* mice with deletion of *Foxp4* in the dental mesenchyme. We discovered that although there was no significant difference in the tooth root length between the molars of control and *Osr2-Cre;Foxp4^{fl/fl}* mice (Fig. 8g–i), larger PDL area and reduced periostin expression was observed largely in the root furcation region in the *Osr2-Cre;Foxp4^{fl/fl}* mice (Fig. 8j–o and Supplementary Fig. 11c), suggesting Foxp4 is required for PDL differentiation during tooth root development, and that GRNs reconstructed in the dental mesenchyme have important function in determining the lineage development during mouse molar morphogenesis.

## Discussion

CNCCs undergo further lineage development after they arrive at their final destinations in different regions of the craniofacial complex. In this study, using the mouse molar as a model, we defined the cellular domains in the dental mesenchyme after postmigratory CNCCs commit to the dental fate and established the transcriptional landscape of these postmigratory CNCCs at single-cell resolution. We show that distinct cellular domains uniquely contribute to tooth development under the control of different transcription factors, highlighting the synchronization of cellular and molecular regulation of lineage development within each domain in the dental mesenchyme. We reveal that cell-cell interaction within the dental mesenchyme is crucial to the development of certain lineages, which highlights the importance of autonomous regulation of postmigratory CNCCs (Fig. 9). Thus, the spatiotemporal data shown in this study serves as an important resource for the study of postmigratory CNCC development and the disorders that result from disruption to this process.

CNCCs receive tremendous interest in the field of developmental biology due to their critical role in building the vertebrate face. Compared to the trunk neural crest, CNCCs have unique features that enable them to give rise to bones and teeth in the craniofacial complex. Several recent studies have investigated the development of cranial bones and sutures as well as the underlying meninges at the single-cell level[39–41] and the data clearly depict the organization of the cellular domains in these cranial crest derived structures, highlighting the progressive establishment of cellular domains by postmigratory CNCCs. CNCCs migrate into the first pharyngeal arch and establish four patterning domains in the mandible along the proximal-distal and oral-aboral axes[42]. Our previous study has identified the cellular heterogeneity within the patterning domains of the mandible and has

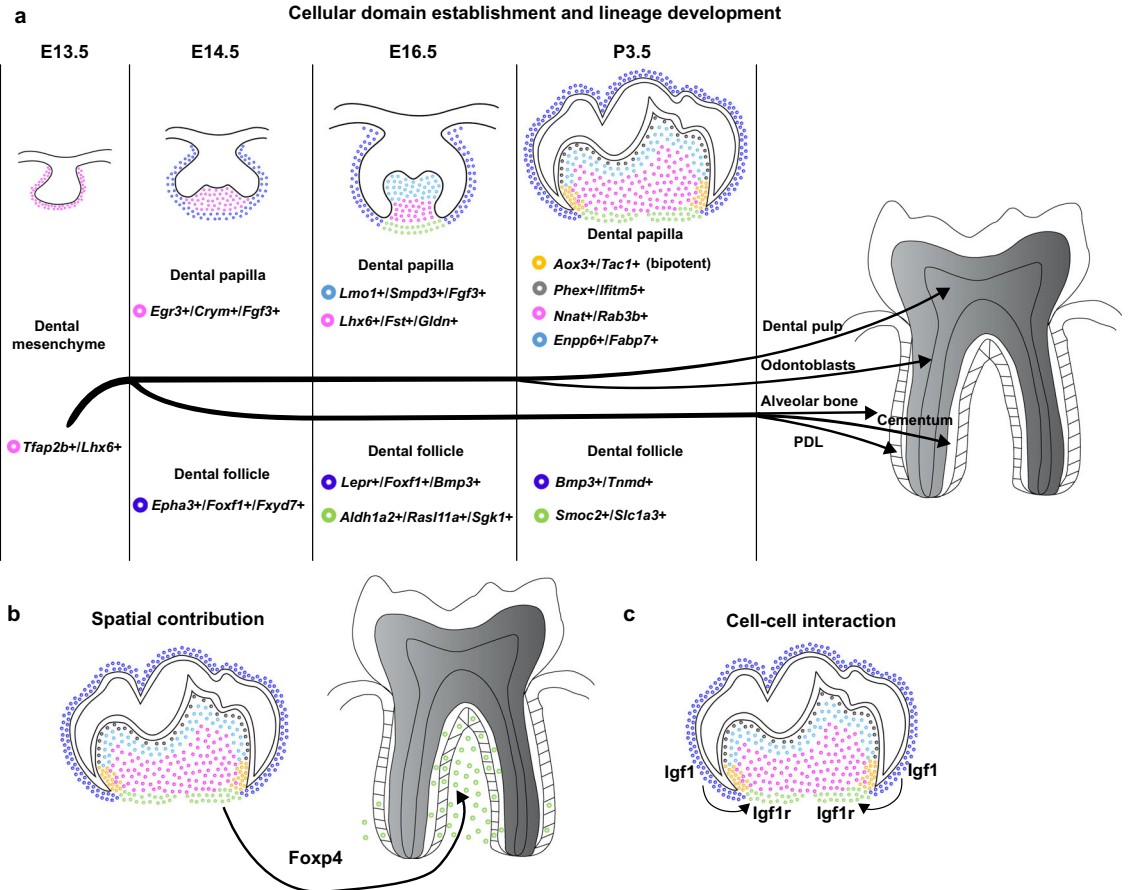

**Fig. 9 | Schematic representation of cell fate decisions in the dental mesenchyme during mouse molar formation. a** Landscape of cellular domain establishment and lineage development in the dental mesenchyme during mouse molar morphogenesis. **b** Apical follicle largely gives rise to periodontal tissue in the furcation region of the mouse molar, which is partially regulated by Foxp4. **c** Igf1-Igf1r signaling mediated cell-cell interaction is indispensable for PDL development.

demonstrated that the cells residing in the proximal domain are progenitors that give rise to major lineages in the mandible, highlighting a spatiotemporal sequence of fate decisions during postmigratory CNCC lineage development[7].

Postmigratory CNCCs undergo cell fate restrictions and ultimately commit to the dental fate after they arrive in the oral domain of the first pharyngeal arch. In this study, we have also uncovered the gradual establishment of the cellular domains in the dental mesenchyme. Our data suggest that the cell populations at E13.5 are relatively homogenous and they maintain multipotency necessary to establish different lineages at later stages. The segregation of dental mesenchymal cells into dental papilla and dental follicle lineages started at E14.5, a first sign of lineage diversification in the dental mesenchyme during molar development. We show numerous different transcription factors enriched in the dental papilla and follicle, and the key determinant for the cell fate choice between these two lineages still needs to be further investigated in the future. Interestingly, the dental follicle cells establish two distinct cellular domains from E16.5, a reflection of different functional deployment in these two cell populations. Indeed, the lineage tracing data from Slc1a3+ cells indicates that the apical follicle domain largely contributes to the periodontal tissue formation in the furcation region of the mouse molar, suggesting that the apical follicle domain governs the lineage development in the root furcation region. Nonetheless, the exact functional difference between these two follicle cellular domains remains to be elucidated in the future.

The cell populations in the dental mesenchyme keep evolving until P3.5, by which time the growth of the tooth root initiates, suggesting that forming the tooth root is more demanding in terms of functional requirements from complex cell populations than forming the crown. However, the mechanisms that maintain the boundaries between different cellular domains in the dental mesenchyme still need to be further explored in the future. Consistent with our findings, the progenitor cells in the dental mesenchyme that contribute to tooth root formation have been shown to be highly heterogeneous[43,44]. Interestingly, we have discovered that apical papilla cells are a uniquely bipotent population that gives rise to both dental pulp and odontoblast lineages at P3.5, providing more comprehensive understanding about the dental papilla lineages beyond what has been previously reported[19]. Collectively, our results reveal that the lineages in the dental mesenchyme are progressively diversified and determined after the commitment of postmigratory CNCCs to the odontogenic fate. The dental mesenchyme in the mouse molar thus provides an excellent model in which to study the lineage development of postmigratory CNCCs.

The development and differentiation of CNCCs are controlled via complex GRNs, comprising transcription factors, cis regulatory elements and epigenetic modifiers[45]. Numerous studies have identified the GRNs in premigratory and migratory CNCCs, expanding our knowledge of the mechanisms that control CNCC development at early stages[46]. Simoes-Costa et al. revealed a cranial neural crest-specific transcriptional circuit which includes Sox8, Tfap2b and Ets1, and found that this circuit not only determines the chondrogenic fate of CNCCs but can also reprogram trunk neural crest cells into craniofacial cartilage, highlighting the power of this GRN to alter cell identity in the neural crest[47]. Early CNCCs maintain patterning plasticity and also respond to local cues to induce positionally specific

transcriptional programs. These distinct transcriptional programs are selectively activated in specific CNCCs but silenced in others. A recent study has shown that positional genes of CNCCs depend on bivalent poised chromatin organization and have accessible H3K27me3/H3K4me2 bivalent distal regulatory elements[48]. The differentiation of CNCCs into distinct derivatives is mediated by gene regulatory programs that compartmentalize these cells into different territories, each with a specific and unique regulatory state. Attanosio et al. have identified the active enhancers in the craniofacial skeleton and their different domains of activity, and showed that craniofacial malformations can arise from deletion of combined enhancer activity[49].

GRNs in the enamel knot of the dental epithelium have been shown to be important for tooth development and patterning. For instance, disruption of Eda signaling leads to a reduced number of cusps and disorganized tooth patterning[50]. In this study, we have identified GRNs in the dental mesenchyme across mouse molar development for the first time. Numerous regulons identified in this study, such as DLX family members Dlx1, Dlx3 and Dlx5, have been shown to play pivotal roles in tooth development. GRNs have also been proposed to be key coordinators during tooth evolution[51]. FOX family genes are also broadly distributed in the identified regulon dataset. FOX transcription factors are evolutionarily conserved across species[52] and they are important for regulating a wide variety of biological functions both during development and in adult organisms. Mutations of FOX genes can have profound effects, seen in human diseases such as Rieger syndrome and lymphedema–distichiasis syndrome[53]. The Foxp family, which is composed of Foxp1, Foxp2, Fxop3 and Foxp4, is also involved in diverse biological processes including development, immune disorders and cancer progression[54]. For example, loss of Foxp1/Foxp2/Foxp4 can lead to disturbed bone and cartilage formation[55]. In this study, we found that Foxp4 is expressed in the dental mesenchyme and is required for the PDL differentiation largely in the root furcation region, highlighting the lineage-specific requirement for root development.

The interaction between different NCC lineages is known to be important for NCC development and is implicated in related diseases[56]. Tissue-tissue interactions between dental mesenchyme and epithelium play a key role in guiding the entire tooth developmental process. However, the cell-cell interaction within the dental mesenchymal linages is poorly understood. The cell-cell interaction network revealed in this study suggests that cell-autonomous regulation within the dental mesenchyme is also important for tooth development. The single-cell atlas established in this study provides a valuable resource for understanding the development of CNCC-derived dental mesenchymal lineages during tooth development. The cellular heterogeneity and domains identified in the dental mesenchyme enable the mouse molar to serve as an excellent model to investigate the lineage development and cell fate decisions of postmigratory CNCCs. The mechanisms uncovered in this study may also have important implications for other CNCC-derived tissues, organs and related diseases.

## Methods
### Animal studies
*Igf1^{flox/flox}*, *Igf1r^{flox/flox}*, *Lepr-Cre*, *Lhx6–CreER^{T2}*, *Rosa2 <fstdTomat>*, and *Slc1a3-CreER^{T2}* mouse strains were purchased from Jackson Laboratory. *Pthrp-CreER^{T2}* mouse line was generously shared by Dr. Wanida Ono[17]. *Foxp4^{flox/flox}* mouse strain was shared by Dr. Edward Morrisey, University of Pennsylvania[57]. *Osr2-Cre* mouse line was shared by Dr. Rulang Jiang, Cincinnati Children's Hospital Medical Center[58]. *Fgf3-CreER^{T2}* mouse line was generously shared by Dr. Hu Zhao, Chinese Institute for Brain Research. A *Pax9-CreER^{T2}* mouse line was generated for this study. Pax9 sgRNA (TCTGATGGGAGCGTCCACTCCGG) was designed to target the 3′ end of Pax9, 34 bp downstream of the target site for the previously generated *Pax9-CreER* mouse[23]. This *Pax9-CreER^{T2}* mouse

line was generated and genotyped as previously described[23], except that 20 ng/μl Cas9 Nuclease v3 and Pax9 sgRNA (0.61pmol/μl crRNA+ 0.61pmol/μl trancrRNA) were used in the injection mix. One male founder mouse was generated for breeding. The tdT reporter mouse line was mated with the *Pax9-CreER^{T2}* line to validate whether the CreER^{T2} activity is comparable to endogenous Pax9 expression. The reagents used for the generation of this *Pax9-CreER^{T2}* line were as below: Alt-R^{TM} S.p. HiFi Cas9 Nuclease v3 (1081060, Integrated DNA Technologies IDT), Alt-R® CRISPR-Cas9 tracrRNA (0000531479, Integrated DNA Technologies IDT) and crRNA (283557335, Integrated DNA Technologies IDT). *Pax9-CreER^{T2}* mouse line will be shared upon request through the corresponding author. All mouse procedures were approved by the Department of Animal Resources and Institutional Animal Care and Use Committee of the University of Southern California (protocol 20299).

All mice were housed under specific-pathogen-free (SPF) conditions in a 12 h light/12 h dark cycle, 18–23 °C and 40–60% humidity. Individuals were identified via ear tags and analyzed in a mixed background. Genotyping was performed as follows: tail biopsies were lysed by incubating at 55 °C overnight in DirectPCR tail solution (Viagen 102T) and heat-inactivated for 30 min at 85 °C for PCR-based genotyping (GoTaq Green MasterMix, Promega, and C1000 Touch Cycler, Bio-rad). Mice were euthanized by $CO_2$ overdose followed by cervical dislocation. All mice were used for analysis regardless of sex. To activate CreER^{T2}, tamoxifen (Sigma-Aldrich) dissolved in corn oil (Sigma-Aldrich, 20 mg/ml) was injected intraperitoneally (200 mg/kg).

### MicroCT analysis
Imaging was performed on a SCANCO μCT50 device at the University of Southern California Molecular Imaging Center with the X-ray source at 70 kVp and 114 μA. The resolution of images was 10 μm. Images were reconstructed in 3D using AVIZO 7.1 (Visualization Sciences Group).

### Immunostaining and RNAscope in situ hybridization
Mouse mandibles were dissected and fixed in 4% PFA (VWR) overnight. After decalcification in 10% EDTA (Sigma-Aldrich) for 4 weeks, samples were incubated in 15% sucrose (VWR) for 2 h followed by 30% sucrose overnight, then embedded in OCT (VWR). Frozen tissue blocks were sectioned at 10 μm on a cryostat (Leica) and mounted on SuperFrost Plus slides (Thermo Fisher Scientific). The tissue sections were blocked for 1 h at room temperature in blocking solution (Vector Laboratories). Sections were then incubated with Rabbit anti-Foxp4 (1:200, HPA007176, MilliporeSigma), Rabbit anti-Periostin (1:100, ab14041, Abcam) and Rabbit anti-Ki67 (1:100, ab15580, Abcam) antibodies diluted in blocking solution at 4 °C overnight. Sections were washed three times with PBS (Thermo Fisher Scientific) prior to incubation with Goat anti-Rabbit secondary antibodies (1:200, A-11011, Thermo Fisher Scientific) in blocking solution at room temperature for 1 h. DAPI (Thermo Fisher Scientific) stained the nuclei. Images were acquired using a Keyence microscope (Carl Zeiss). Non-immune immunoglobulins of the same isotype as the primary antibodies were used as negative controls.

ISH was performed using an RNAscope Multiplex Fluorescent Assay (Advanced Cell Diagnostics). Briefly, tissues were fixed in 4% PFA overnight at room temperature before cryosectioning to 10 μm thickness. Target retrieval was performed on the sections for 5 min at 95–100 °C, followed by protease (Advanced Cell Diagnostics) treatment for 7 min at 40 °C. Probes (Advanced Cell Diagnostics) were then hybridized for 2 h at 40 °C followed by RNAscope amplification reagents provided in RNAscope Multiplex Fluorescent Reagent Kit v2 (Advanced Cell Diagnostics). The signal was detected by TSA Plus Cyanine 3 (PerkinElmer).

### EdU incorporation and staining

Mouse pups at P3.5 were injected intraperitoneally with EdU (Thermo Fisher Scientific, 10 μg/g body weight) before being euthanized. Mandibles were then fixed and decalcified as described above. Click-iT plus EdU cell proliferation kit (Thermo Fisher Scientific, C10637) was used to detect EdU according to the manufacturer's instructions.

### ImageJ analysis

ImageJ was used to determine the tooth root length and PDL area. Images of the controls and mutants were first converted to 8-bit binary. The root length and PDL area were then calculated by the Analyze function.

### Statistics and reproducibility

GraphPad Prism 9 was used to perform all statistical analyses and the data are presented as mean ± standard deviation unless otherwise stated. Two-tailed Student's $t$ test or one-way ANOVA were used for comparisons, with $P < 0.05$ considered statistically significant. All immunostaining, RNAscope staining and lineage tracing experiments were performed with at least three experimental replicates.

### Single-cell RNA sequencing

**Isolation of cells and sequencing.** Single-cell transcriptomes were obtained from the digestion of the tooth and its surrounding tissue at different stages (E13.5, E14.5, E16.5, P3.5, and P7.5). In brief, the tooth and its surrounding tissue were placed in 4 mg/ml Dispase (Roche) and 2 mg/ml Collagenase I (Gibco) on a thermomixer (Eppendorf) at 37 °C for 15 to 30 min depending on the stage of the sample to release cells from the tissue. All our samples had more than 70% viable cells. For each sample, we targeted 20,000 cells for scRNA sequencing. The numbers of actually sequenced cells were 21,416 cells for E13.5, 26,461 cells for E14.5, 29,766 cells for E16.5, 19,134 cells for P3.5, and 15,462 cells for P7.5. On average, we had a sequencing depth of 80,000 read pairs per cell. At each stage, we used two biological replicates for sequencing analysis. Quality control, mapping, and count table assembly of the library were performed using the CellRanger pipeline version 3.1.0.

**Identifying variable genes and dimensionality reduction.** Raw read counts from the cells at each stage were analyzed using the Seurat 4.0R package[59]. Low gene expression cells were filtered out following standard Seurat object generation. Poor-quality cells with less than 200 genes and mitochondrial gene percentages >50 were removed. Sctransform was applied for normalization and cell cycle regression. RunPCA and RunUMAP were performed for dimensionality reduction and final visualization of the clustering.

**Subcluster analysis.** Subcluster analysis was performed to investigate the heterogeneity within the dental mesenchymal populations. Published markers for the dental mesenchyme identified the dental mesenchymal cell populations in the mouse molar.

**Integrative analysis of samples at different stages.** Seurat 4 was used to combine the single-cell data from five stages and perform integration analysis. The PrepSCTIntegration function was performed before identifying anchors with the function FindIntegrationAnchors. Seurat objects were then returned by passing these anchors to the IntegrateData function. RunPCA and RunUMAP visualization were used for downstream analysis and visualization.

**Monocle trajectory analysis.** We used Monocle 3 to generate the pseudotime trajectory across the dental mesenchymal cells. The cells were input to Monocle to infer cluster and lineage relationships within a given cell type. UMAP embeddings and cell subclusters generated from Seurat were converted to a cell_data_set object using SeuratWrappers (v.0.1.0) and then trajectory graph learning and pseudotime measurement were performed through reversed graph embedding. The estimation of the root node of the trajectory was based on the early dental mesenchyme marker Tfap2b.

**GeneSwitches analysis.** Following Monocle trajectory analysis, pseudo-times were subsetted using choose_graph_segments() and expression plots were generated using plot_cells(). Analysis of the genes switching on and off in pseudo-time was performed using the GeneSwitches package.

**Velocity analysis.** Python-based Velocyto command-line tool and Velocyto R package were used as instructed for RNA velocity analysis. Velocyto calculated the single-cell trajectory/directionality using spliced and unspliced reads. From loom files generated by the command-line tool, we extracted the dental papilla cells in R to perform the Velocity analysis using the SeuratWrappers package. RNA velocity was estimated using a gene-relative model with kNN cell pooling ($k = 25$). RNA velocity was visualized on the UMAP embedding with the parameter $n = 200$. scVelo was used to perform Velocity analysis of integrative datasets combining different stages.

**Gene regulatory network inference.** To better understand the transcription factors activating gene expression in the dental mesenchymal cells, we utilized the R package SCENIC, which is a computational method to detect gene regulatory networks. The network activity was analyzed in the dental mesenchymal cells to identify recurrent cellular states. Transcription factors were identified using GENIE3 and compiled into regulons, then subjected to cis-regulatory motif analysis. Regulon activity was then scored using AUCell.

**Ligand–receptor interaction analysis.** CellChat was performed to identify the potential ligand-receptor interactions in the dental mesenchymal cells. To infer potential cell–cell communication networks, we followed the standard workflow, imported the Seurat object as the input into CellChat and used preprocessing functions identifyOverExpressedGenes, identifyOverExpressedInteractions and projectData with standard parameters set. We then ran the core functions computeCommunProb, computeCommunProbPathway and aggregateNet with standard parameters. Finally, the functions netVisual_circle, netVisual_bubble, netAnalysis_signalingRole_heatmap and netAnalysis_signalingRole_network were applied to the network to determine the senders and receivers.

### Reporting summary

Further information on research design is available in the Nature Research Reporting Summary linked to this article.

## Data availability

All single-cell RNA-seq datasets are available through the GEO database under accession code "GSE189381" and FaceBase data repository (Record ID A-QVNG, https://doi.org/10.25550/A-QVNG). All other relevant data supporting the key findings of this study are available within the article and its Supplementary Information files or from the corresponding author upon reasonable request. Source data are provided with this paper.

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

## Acknowledgements

We thank Bridget Samuels for critical reading of the manuscript. We thank Dr. Hu Zhao and Yuling Wang from the Chinese Institute for Brain Research for collecting samples from *Fgf3-CreER^{T2};tdT* mice. We acknowledge USC Libraries Bioinformatics Service for assisting with data analysis. The bioinformatics software and computing resources used in the analysis are funded by the USC Office of Research and the Norris Medical Library. This study was supported by grants from the National Institute of Dental and Craniofacial Research, National Institutes of Health (R01 DE022503 to Y.C. and R01 DE012711 to Y.C.).

## Author contributions

J.J. and Y.C. designed the study. J.J. carried out most of the experiments and analyzed the data. J.F., Y.Y., T.G., J.L., and F.P. participated in the sample collection. J.F. participated in the generation of *Pax9-CreER^{T2}* mouse line. T-V.H. participated in the microCT analysis. J.J. and Y.C. co-wrote the paper. Y.C. supervised the research.

## Competing interests

The authors declare no competing interests.
