## [Peer Review File · Nature Communications]

REVIEWER COMMENTS

Reviewer #1 (Remarks to the Author):

This manuscript explores differentiation of post-migratory cranial neural crest cells during tooth morphogenesis using single-cell RNAseq across multiple timepoints. The authors identify a gene regulatory network in the dental mesenchyme contributing to tooth development and pinpoint novel markers of specific cellular domains across time. Further, they highlight potential intercellular signaling networks that contribute to tooth morphogenesis. The results will be of particular interest to investigators studying tooth development, as well as investigators studying lineage commitment of cranial neural crest cells. The manuscript would benefit from a more thorough (re)interpretation of lineage tracing experiments, a demonstration of the lineage trajectory from apical to middle to coronal papilla, and a more thorough demonstration of compromised PDL formation in the various conditional knockout mouse models.

Major comments:

1. In Figures 3 and 4, the tdT pattern arising in animals expressing *Lepr-Cre;tdT* (marking lateral follicle cells) and *Slc1a3-CreER;tdT* (marking apical follicle cells) at P21 appear very similar, though the authors state that the latter preferentially contributes to the periodontal tissue in the root furcation region. The authors should more clearly indicate the root furcation region in Figure 3S and Figure 4N-O and temper statements related to preferential contribution of the apical follicle domain to this region.
2. In Figure 5C and 5D, cells outside of the apical papilla are also Ki67-positive at P3.5 and EdU-positive at P5.5. The authors cannot rule out that these cycling cells are contributing to the odontoblast and pulp lineages at P12.5. Similarly, beyond its expression in the apical papilla, *Axin2* is also expressed in what appears to be the middle papilla and/or apical follicle in Figure 5G. The authors cannot rule out from their lineage tracing experiments that these are the cells contributing to the dental pulp and odontoblasts. Given the more specific expression patterns of *Aox3* and *Tac1* in the apical papilla in Figure S5N and S5O, these would have been more suitable drivers for lineage tracing studies of the apical papilla.
3. In Figure 5A (middle), it appears as though the trajectory from apical papilla to coronal papilla has a much stronger velocity than the trajectory from apical to middle to coronal papilla. The authors need to more thoroughly demonstrate this lineage trajectory in order to make this claim in Figure 5A (right) and 5J.
4. It is unclear why the authors used the *Gli1-CreERT2* driver for the experiments in Figure 7, given their previous use of the *Lepr-Cre* driver to mark lateral follicle cells. Was *Gli1* detected as a transcript enriched in the lateral follicle in the scRNAseq data? It does not appear in Figure 3B or 4B. At a

minimum, the authors need to cite previous references demonstrating activity of this driver specifically in lateral follicle cells.

5. It is unclear from the histology images in Figure 7J, M and P how PDL formation is compromised. The authors should indicate in the figure any histological defects. Is periostin a marker of PDL formation or differentiation in Figure 7K, N and Q? If the latter, differentiation and not formation may be affected. Root length should be quantified upon conditional loss of *Igf1* and *Igf1r* to rule out a general delay in tooth development in these mutant animals. Given that the only defect in tooth development detected upon conditional loss of *Igf1* and *Igf1r* is reduced periostin staining, statements such as “We found that cell-cell interaction within the dental mesenchyme is crucial to tooth development” should be tempered.

6. Similar to the comments above, is periostin serving as a marker of PDL differentiation in Figure 8N and O? Given the reduced root length upon conditional loss of *Foxp4*, could the decreased expression of periostin stem from a general delay in tooth development? Again, given the phenotype in these animals, the phrase “...*Foxp4* is expressed in the dental mesenchyme and is required for the PDL formation preferentially in the root furcation region...” should be tempered.

Minor comments:

1. The authors should label the dental pulp and odontoblast layers in Figure 5E and 5I.
2. Can the authors highlight *Enpp6* in Figure 6K and *Aldh1a2* in Figure 6U?

Reviewer #2 (Remarks to the Author):

The manuscript “Spatiotemporal single-cell regulatory atlas reveals neural crest lineage diversification and cellular function during tooth morphogenesis” takes advantage of single-cell transcriptomics and uses this technique in the identification of mouse molar development. The authors present a high-quality and very complex manuscript and focus on the formation of the dental pulp and the follicle, which are both of cranial neural crest origin. In the manuscript are presented data from sequenced molars during different stages of development and most of the data are validated in vivo by in situ hybridization and, importantly by the lineage tracing, which is highly appreciated. Moreover, the authors produced floxed mice and performed several functional studies which further confirm their findings. Although the manuscript is of generally high quality, I have several questions and suggestions.

The introduction is very well written but focuses mostly on the cranial neural crest however the manuscript itself is focused purely on molar development. It would be appropriate to refer more to teeth/molars.

Plots showing the expression of specific genes are produced only from the isolated mesenchyme. Better will be to show the expression of selected marker genes also in the whole dataset in the supplementary

figures to refer better to the situation in vivo (showed by IHC) where also other populations are represented.

In Fig. 1B,C,E what is the population that is Tfp2b negative (on the left side)? It is not explained nor validated. What is shown in Fig 1B?

The lineage tracing of Lhx6 is missing timepoint P21, which is shown in the Pax9 tracing. What is the reason?

Slc1a3 lineage tracing shows numerous traced cells in the alveolar bone but from IHC is not apparent other positive staining except of the apical follicle. Do the apical follicle cells contribute to bone formation? This needs to be explained.

The differentiation trajectory in the Fig. 5A (very right) and then Fig. 5J is misinterpreted. It cannot be claimed from the RNA Velocity (5A middle) that the differentiation goes through the middle papilla and there are no other presented data that would prove this.

The generally accepted knowledge is that odontoblasts are long-lasting cells that are not replaced in healthy tissue. The Axin2 lineage tracing (Fig. 5G-I) shows that Axin2+ cells traced at P3.5 give rise to almost all odontoblasts and many pulp cells in the crown in P21.5. However, in the P5.5 no odontoblasts are traced. This is a disturbing observation and barely possible situation that needs to be explained/repeated.

In the Fig. 6 showing RNA velocity will help to support observed findings. In the Fig. 6A and 6L are only 2 colors (timepoints), where are other timepoints? In the legend are shown all of them.

The figure 7I-Q would greatly benefit by showing uCT analyses of impaired molars with a special focus on dentin – there is a possible dentin phenotype in the Gli1CreERT2;Igf1flfl mice.

Minor points:

Missing information of the amount of injected tamoxifen in different stages, only the concentration is stated.

Typo in methods: “10 mm sections”, should be 10 μ m.

June 22, 2022

NCOMMS-21-46531

Point-by-point response to the reviewers' comments:

Reviewer #1 (Remarks to the Author):

This manuscript explores differentiation of post-migratory cranial neural crest cells during tooth morphogenesis using single-cell RNAseq across multiple time points. The authors identify a gene regulatory network in the dental mesenchyme contributing to tooth development and pinpoint novel markers of specific cellular domains across time. Further, they highlight potential intercellular signaling networks that contribute to tooth morphogenesis. The results will be of particular interest to investigators studying tooth development, as well as investigators studying lineage commitment of cranial neural crest cells. The manuscript would benefit from a more thorough (re)interpretation of lineage tracing experiments, a demonstration of the lineage trajectory from apical to middle to coronal papilla, and a more thorough demonstration of compromised PDL formation in the various conditional knockout mouse models.

Major comments:

1. In Figures 3 and 4, the tdT pattern arising in animals expressing *Lepr-Cre;tdT* (marking lateral follicle cells) and *Slc1a3-CreER;tdT* (marking apical follicle cells) at P21 appear very similar, though the authors state that the latter preferentially contributes to the periodontal tissue in the root furcation region. The authors should more clearly indicate the root furcation region in Figure 3S and Figure 4N-O and temper statements related to preferential contribution of the apical follicle domain to this region.

We thank the reviewer for this suggestion. The root furcation region is where the roots are separated in the multi-rooted mouse molar[1]. We have added this statement in the manuscript. We have used arrows with different colors to indicate the PDL (yellow) and alveolar bone (green) in the root furcation region in Fig. 3s and Fig. 4o. We agree that *Lepr+* cells at E16.5 in the lateral follicle can also contribute to the periodontal tissue formation in the root furcation region. Indeed, the functional difference between the lateral and apical follicle in the mouse molar needs to be further studied in the future. To reflect this, we have changed the title for this section in the Results to “Evolution and spatial separation of cellular domains in the mouse molar” on page 6 and added the statement “Nonetheless, the exact

functional difference between these two follicle cellular domains remains to be elucidated in the future” in the Discussion on page 12 of the manuscript.

2. In Figure 5C and 5D, cells outside of the apical papilla are also Ki67-positive at P3.5 and EdU-positive at P5.5. The authors cannot rule out that these cycling cells are contributing to the odontoblast and pulp lineages at P12.5. Similarly, beyond its expression in the apical papilla, Axin2 is also expressed in what appears to be the middle papilla and/or apical follicle in Figure 5G. The authors cannot rule out from their lineage tracing experiments that these are the cells contributing to the dental pulp and odontoblasts. Given the more specific expression patterns of Aox3 and Tac1 in the apical papilla in Figure S5N and S5O, these would have been more suitable drivers for lineage tracing studies of the apical papilla.

We appreciate the reviewer for this insight. We agree that both cycling cells and Axin2+ cells are not specifically restricted in the apical papilla region of mouse molars at P3.5. Therefore, we cannot rule out the possibility that cells surrounding the apical papilla can also give rise to dental pulp and odontoblasts. Unfortunately, inducible Cre lines for *Aox3* and *Tac1* are not available. We have further identified other specific markers for apical papilla and found that *Fgf3* is also specifically expressed in the apical papilla in the mouse molar at P3.5 (new Fig. 5f, g). Therefore, we have generated *Fgf3-CreER^{T2};tdT* mice to perform the lineage tracing experiments. We found that *Fgf3*+ cells in the apical papilla can give rise to odontoblasts and dental pulp cells in the mouse molar, suggesting that cell population in the apical papilla is the bipotent progenitor population for odontoblast and dental pulp lineages (new Fig. 5h, i). We have replaced original Fig. 5f-i with *Fgf3* expression data and lineage tracing analysis from *Fgf3-CreER^{T2};tdT* mice in the manuscript.

3. In Figure 5A (middle), it appears as though the trajectory from apical papilla to coronal papilla has a much stronger velocity than the trajectory from apical to middle to coronal papilla. The authors need to more thoroughly demonstrate this lineage trajectory in order to make this claim in Figure 5A (right) and 5J.

We appreciate the reviewer for this suggestion. We agree it is possible that apical papilla can either become coronal papilla directly or through the intermediate middle papilla status. We have modified the route of the trajectory to reflect the possible path based on the velocity analysis (Fig. 5a and Fig. 5j). Indeed, the exact *in vivo* differentiation trajectory of the dental papilla population needs to be further examined in the future.

4. It is unclear why the authors used the *Gli1-CreERT2* driver for the experiments in Figure 7, given their previous use of the *Lepr-Cre* driver to mark lateral follicle cells. Was *Gli1* detected as a transcript enriched in the lateral follicle in the scRNAseq data? It does not appear in Figure 3B or 4B. At a minimum, the authors need to cite previous references demonstrating activity of this driver specifically in lateral follicle cells.

We apologize for this confusion. *Gli1*⁺ cells were previously identified as a progenitor cell population giving rise to both dental follicle and dental papilla in molar root development[2]. We used the *Gli1-CreERT2* line because it is efficient in deleting the genes in the dental follicle. To more specifically delete *Igf1* in the lateral follicle, we have further generated *Lepr-Cre;Igf1^{fl/fl}* mice and found these mutant mice showed a similar PDL phenotype to that observed in *Gli1-CreERT2;Igf1^{fl/fl}* mice. We have replaced *Gli1-CreERT2;Igf1^{fl/fl}* with this *Lepr-Cre;Igf1^{fl/fl}* model in Fig. 7i-n in the revised manuscript.

5. It is unclear from the histology images in Figure 7J, M and P how PDL formation is compromised. The authors should indicate in the figure any histological defects. Is periostin a marker of PDL formation or differentiation in Figure 7K, N and Q? If the latter, differentiation and not formation may be affected. Root length should be quantified upon conditional loss of *Igf1* and *Igf1r* to rule out a general delay in tooth development in these mutant animals. Given that the only defect in tooth development detected upon conditional loss of *Igf1* and *Igf1r* is reduced periostin staining, statements such as “We found that cell-cell interaction within the dental mesenchyme is crucial to tooth development” should be tempered.

We thank the reviewer for this comment. We have outlined and measured the area of the PDL based on the H&E staining in Fig. 7 and found that the PDL area is larger in the mutants than controls, suggesting relatively undifferentiated status of PDL in the mutants. Periostin is a marker for PDL differentiation; therefore, it is indeed the differentiation rather than formation of PDL that is affected after the disruption of *Igf1* signaling in the mouse molar. We also measured the root length of mouse molars at an earlier time point (P16.5) when there is no significant difference between the controls and mutants, and we found that the periostin expression is already reduced in the mutants, suggesting the defect is not due to delayed tooth development (shown in Fig. 1 in this letter). We have added this information to Fig. 7i-q and Supplementary Fig. 11a, b in the manuscript. We have changed the corresponding section title in the Results to “Cell-cell interaction between the cellular

domains in the dental mesenchyme is important for lineage development in the mouse molar” on page 9 and modified statements throughout the manuscript to highlight PDL lineage differentiation.

Fig. 1 Igf1 signaling mediated cell-cell interaction is important for PDL differentiation in the mouse molar. **a-i** H&E staining and immunostaining of periostin in the mouse molar in control, *Lepr-Cre;Igf1^{fl/fl}* and *Slc1a3-CreER^{T2};Igf1^{fl/fl}* mice at P16.5. **j-k** quantitative analysis of root length and PDL area in control, *Lepr-Cre;Igf1^{fl/fl}* and *Slc1a3-CreER^{T2};Igf1^{fl/fl}* molars at P16.5.

6. Similar to the comments above, is periostin serving as a marker of PDL differentiation in Figure 8N and O? Given the reduced root length upon conditional loss of *Foxp4*, could the decreased expression of periostin stem from a general delay in tooth development? Again, given the phenotype in these animals, the phrase “...*Foxp4* is expressed in the dental mesenchyme and is required for the PDL formation preferentially in the root furcation region...” should be tempered.

We appreciate the comment from the reviewer. As mentioned above, periostin is a marker for PDL differentiation. In order to rule out the possibility that the reduced periostin expression was due to delayed tooth development, we collected the samples at an earlier time

point when the root length in the control and mutant molars is still the same, namely P16.5. We found that the expression of periostin was already reduced in the PDL of *Osr2-Cre;Foxp4^{fl/fl}* mice at this stage, suggesting that the defect in PDL is not owing to the delayed tooth development (shown in Fig. 2 in this letter). We have added this to Fig. 8g-o and Supplementary Fig. 11c in the manuscript, and also modified statements throughout the manuscript to highlight the PDL differentiation defect in *Foxp4* mutant model.

Fig. 2 Loss of *Foxp4* in the dental mesenchyme leads to compromised PDL differentiation in the mouse molar. a-c CT scanning and quantitative analysis of the tooth root length in *Foxp4^{fl/fl}* and *Osr2-Cre;Foxp4^{fl/fl}* molars. d-i H&E staining and immunostaining of periostin in the molars of *Foxp4^{fl/fl}* and *Osr2-Cre;Foxp4^{fl/fl}* mice at P16.5. j quantitative analysis of PDL area in *Foxp4^{fl/fl}* and *Osr2-Cre;Foxp4^{fl/fl}* molars at P16.5. Black and white dotted lines outline the PDL in the mouse molar. The white arrows point to the positive signals and the asterisks indicate absence of the signal.

Minor comments:

1. The authors should label the dental pulp and odontoblast layers in Figure 5E and 5I.

We have labeled the dental pulp and odontoblast layers in Fig. 5e and 5i.

2. Can the authors highlight *Enpp6* in Figure 6K and *Aldh1a2* in Figure 6U?

We have highlighted *Enpp6* in Fig. 6k and *Aldh1a2* in Fig. 6u with red arrows.

Reviewer #2 (Remarks to the Author):

The manuscript “Spatiotemporal single-cell regulatory atlas reveals neural crest lineage diversification and cellular function during tooth morphogenesis” takes advantage of single-cell transcriptomics and uses this technique in the identification of mouse molar development. The authors present a high-quality and very complex manuscript and focus on the formation of the dental pulp and the follicle, which are both of cranial neural crest origin. In the manuscript are presented data from sequenced molars during different stages of development and most of the data

are validated in vivo by in situ hybridization and, importantly by the lineage tracing, which is highly appreciated. Moreover, the authors produced floxed mice and performed several functional studies which further confirm their findings. Although the manuscript is of generally high quality, I have several questions and suggestions.

1. The introduction is very well written but focuses mostly on the cranial neural crest however the manuscript itself is focused purely on molar development. It would be appropriate to refer more to teeth/molars.

We thank the reviewer for this suggestion. We have changed the introduction accordingly to emphasize tooth development and the rationale behind using the tooth as a model to study postmigratory CNCCs on page 4 in this manuscript.

2. Plots showing the expression of specific genes are produced only from the isolated mesenchyme. Better will be to show the expression of selected marker genes also in the whole dataset in the supplementary figures to refer better to the situation in vivo (showed by IHC) where also other populations are represented.

We appreciate the reviewer for this suggestion. We have plotted the marker genes in the whole dataset to refer to the other cell populations within the mouse molar and its surrounding tissues. We have added the plots as new Supplementary Fig. 1, 2, 4 and 6.

3. In Fig. 1B, C, E what is the population that is *Tfap2b* negative (on the left side)? It is not explained nor validated. What is shown in Fig 1B?

We thank the reviewer for pointing this out. Compared with *Lhx6*⁺ cells, *Tfap2b*⁺ cells only partially represent the dental mesenchymal population at E13.5; therefore, the *Tfap2b* negative population in Fig. 1c are the cells that express *Lhx6* but don't express *Tfap2b* at this stage. Fig. 1b is the whole population of the dental mesenchyme of the mouse molar at E13.5. We have used blue color to display the dots in Fig. 1b to avoid confusion.

4. The lineage tracing of *Lhx6* is missing time point P21, which is shown in the *Pax9* tracing. What is the reason?

We thank the reviewer for this insight. We have provided lineage tracing data from *Lhx6-CreER;tdT* mice at several more time points including P21.5 in Fig. 1h-m of the revised manuscript.

5. *Slc1a3* lineage tracing shows numerous traced cells in the alveolar bone but from IHC is not

apparent other positive staining except of the apical follicle. Do the apical follicle cells contribute to bone formation? This needs to be explained.

We thank the reviewer for this comment. Dental follicle cells harbor the progenitor cells that can give rise to periodontal tissue such as PDL and alveolar bone[3]. Slc1a3+ apical follicle cells indeed contribute to the alveolar bone formation based on the lineage tracing data in the manuscript (Fig. 4m, o). We have added a description in the manuscript on page 7.

6. The differentiation trajectory in the Fig. 5A (very right) and then Fig. 5J is misinterpreted. It cannot be claimed from the RNA Velocity (5A middle) that the differentiation goes through the middle papilla and there are no other presented data that would prove this.

We appreciate the reviewer for this comment. We agree that the *in vivo* differentiation path in the dental papilla needs to be further investigated in the future. Based on the Velocity signal, we proposed two routes: the apical papilla can either become coronal papilla directly or through the middle papilla. We have modified the differentiation path to reflect the possible differentiation routes in Fig. 5a and Fig. 5j in the manuscript.

7. The generally accepted knowledge is that odontoblasts are long-lasting cells that are not replaced in healthy tissue. The Axin2 lineage tracing (Fig. 5G-I) shows that Axin2+ cells traced at P3.5 give rise to almost all odontoblasts and many pulp cells in the crown in P21.5. However, in the P5.5 no odontoblasts are traced. This is a disturbing observation and barely possible situation that needs to be explained/repeated.

We thank the reviewer for this comment. We agree that odontoblasts have very slow turnover and are long-lasting after tooth development is complete. However, the odontoblasts are being replenished during the tooth development process[4]. We have also performed lineage tracing analysis for several more time points in *Axin2-CreER^{T2};tdT* mice between P5.5 and P21.5. The data suggest that Axin2+ cells indeed contribute to most of the odontoblast formation during the tooth development (shown in Fig. 3 in this letter). We have removed this data from the manuscript because *Axin2* is not a specific marker for apical papilla at P3.5 and we have now instead used *Fgf3* according to the suggestions from reviewer #1.

Fig. 3 Axin2⁺ cell contribute to odontoblasts and dental pulp cells during root development of the mouse molar. **a-h** lineage tracing analysis of Axin2⁺ cells in the mouse molar of *Axin2-CreER^{T2};tdT* mice at different time points after the induction of tamoxifen. Arrows indicate positive signals and the asterisks indicate absence of the signal.

8. In the Fig. 6 showing RNA velocity will help to support observed findings. In the Fig. 6A and 6L are only 2 colors (time points), where are other time points? In the legend are shown all of them. **We appreciate the reviewer for this insight. We have also performed RNA velocity analysis for dental papilla and follicle lineages and the outcome is consistent with Monocle analysis. We have added the velocity data to Fig. 6c and Fig. 6n in the manuscript. Cell populations from embryonic stages including E13.5, E14.5 and E16.5 are largely present in the progenitor cell group, consistent with their relatively undifferentiated status. We have highlighted the progenitor group with higher magnification to confirm this (shown in Fig. 4 in this letter).**

Fig. 4 Integrative analysis of the dental mesenchymal lineage data from E13.5, E14.5, E16.5, P3.5 and P7.5. a-b Integrative analysis of the dental papilla lineage. **c** Velocity analysis of the dental papilla lineage. **d-e** Integrative analysis of the dental follicle lineage. **f** Velocity analysis of the dental follicle lineage. Black arrows indicate the differentiation trajectory.

9. The figure 7I-Q would greatly benefit by showing uCT analyses of impaired molars with a special focus on dentin – there is a possible dentin phenotype in the *Gli1-Cre^{ERT2};Igf1^{fl/fl}* mice.

We thank the reviewer for this comment. We have collected more samples and conducted microCT analysis of the controls and mutants and found there is no apparent dentin phenotype after the Igf1 signaling is disrupted. We have confirmed this with more representative images of H&E and *Dspp* staining which indicate the differentiation status of odontoblasts at P18.5 (shown in Fig. 5 in this letter). According to the suggestion from reviewer #1, we have generated *Lepr-Cre;Igf1^{fl/fl}* models which specifically delete *Igf1* in the lateral dental follicle of the mouse molar. The phenotype of *Lepr-Cre;Igf1^{fl/fl}* models is similar to that of *Gli1-Cre^{ERT2};Igf1^{fl/fl}* mice, suggesting the indispensable role of Igf1 signaling in the PDL differentiation.

Fig. 5 Disruption of *Igf1* signaling has no effect on odontoblast differentiation. **a-l** CT scanning, H&E staining and RNAscope staining of *Dspp* in the molars of Control, *Gli1-Cre^{ERT2};Igf1^{fl/fl}* and *Slc1a3-Cre^{ERT2};Igf1^{fl/fl}* mice.

Minor points:

1. Missing information of the amount of injected tamoxifen in different stages, only the concentration is stated.

We thank the reviewer for this suggestion. We have added the information about the amount of injected tamoxifen at different stages.

2. Typo in methods: “10 mm sections”, should be 10 μ m.

We thank the reviewer for this insight. We have corrected this typo.

We greatly appreciate all of the insightful suggestions offered by the reviewers and feel that our revised manuscript has greatly improved through this review process. Thank you very much.

Sincerely,

Yang Chai, DDS, PhD
University Professor
CCMB, USC

References

1. He J, Jing J, Feng J, Han X, Yuan Y, Guo T, Pei F, Ma Y, Cho C, Ho TV, Chai Y. Lhx6 regulates canonical Wnt signaling to control the fate of mesenchymal progenitor cells during mouse molar root patterning. *PLoS Genet.* 2021,17(2):e1009320.
2. Liu Y, Feng J, Li J, Zhao H, Ho TV, Chai Y. An Nfic-hedgehog signaling cascade regulates tooth root development. *Development.* 2015, 142(19):3374-82
3. Li J, Parada C, Chai Y. Cellular and molecular mechanisms of tooth root development. *Development.* 2017, 144(3):374-384.
4. Du J, Jing J, Yuan Y, Feng J, Han X, Chen S, Li X, Peng W, Xu J, Ho TV, Jiang X, Chai Y. Arid1a-Plagl1-Hh signaling is indispensable for differentiation-associated cell cycle arrest of tooth root progenitors. *Cell Rep.* 2021, 35(1):108964.

REVIEWERS' COMMENTS

Reviewer #1 (Remarks to the Author):

The authors have addressed my original concerns, both editorially and through the introduction of new experimental results.

Reviewer #2 (Remarks to the Author):

The authors have addressed all my questions/comments and have now provided a significantly improved revision of the manuscript, which I am happy to recommend for publication.

I have no further comments.

1 July 22, 2022
2 NCOMMS-21-46531

3

4 Reviewer #1 (Remarks to the Author):

5 The authors have addressed my original concerns, both editorially and through the introduction of
6 new experimental results.

7 **We thank the reviewer for the helpful comments.**

8

9 Reviewer #2 (Remarks to the Author):

10 The authors have addressed all my questions/comments and have now provided a significantly
11 improved revision of the manuscript, which I am happy to recommend for publication. I have no
12 further comments.

13 **We appreciate this reviewer for the suggestions and comments.**